# SoftCFG: Uncertainty-guided Stable Guidance for Visual autoregressive Model

**Dongli Xu**[1][*] **Aleksei Tiulpin**[2,3] **, Matthew B. Blaschko**[1]
[1]KU Leuven [2]University of Oulu [3]Weill Cornell Medicine

## Abstract

Autoregressive (AR) models have emerged as powerful tools for image generation by modeling images as sequences of discrete tokens. While Classifier-Free Guidance (CFG) has been adopted to improve conditional generation, its application in AR models faces two key issues: *guidance diminishing*, where the conditional–unconditional gap quickly vanishes as decoding progresses, and *over-guidance*, where strong conditions distort visual coherence. To address these challenges, we propose **SoftCFG**, an uncertainty-guided inference method that distributes adaptive perturbations across all tokens in the sequence. *The key idea behind SoftCFG* is to let each generated token contribute certainty-weighted guidance, ensuring that the signal persists across steps while resolving conflicts between text guidance and visual context. To further stabilize long-sequence generation, we introduce **Step Normalization**, which bounds cumulative perturbations of SoftCFG. Our method is training-free, model-agnostic, and seamlessly integrates with existing AR pipelines. Experiments show that SoftCFG significantly improves image quality over standard CFG and achieves state-of-the-art FID on ImageNet $256 \times 256$ among autoregressive models. Codebase is available.

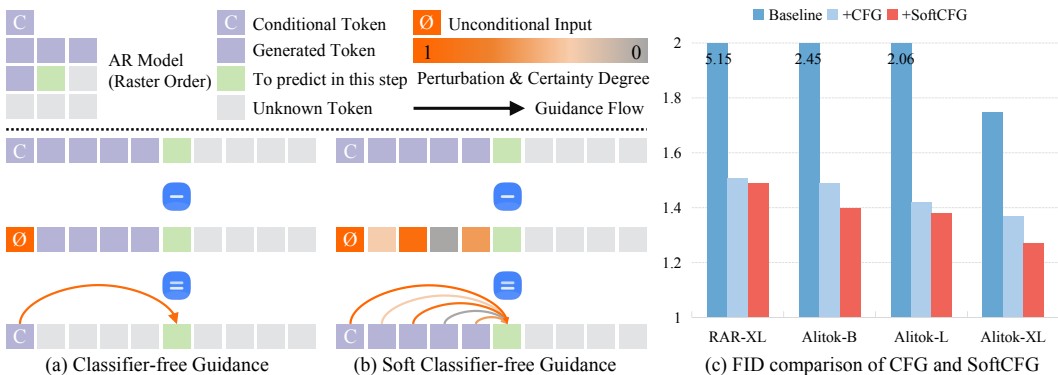

Figure 1: Comparison of standard CFG and SoftCFG. (a) Standard CFG for AR replaces the first class token with an empty token. (b) SoftCFG scales the perturbation by model uncertainty, yielding smoother, more informative guidance. (c) On ImageNet-$256 \times 256$, SoftCFG consistently improves FID across diverse AR models vs. baseline sampling and standard CFG.

## 1 Introduction

Visual Autoregressive (AR) models (Van den Oord et al., 2016; Chang et al., 2022; Sun et al., 2024) formulate image generation as a next-token prediction task over sequences of discrete visual tokens, typically obtained via vector-quantized autoencoders (Van Den Oord et al., 2017). Given a sequence of previously generated tokens, a visual AR model predicts the next one with a decoder-only transformer, following the same autoregressive formulation that underlies large language models (LLMs) (Touvron et al., 2023). This unified design enables conceptual and architectural alignment between vision and language, offering simplicity, scalability, and potential for cross-modal integration (Wu et al., 2025a; Xie et al., 2025; Xin et al., 2025).

---

[*]Corresponding Author: dongliixu@gmail.com

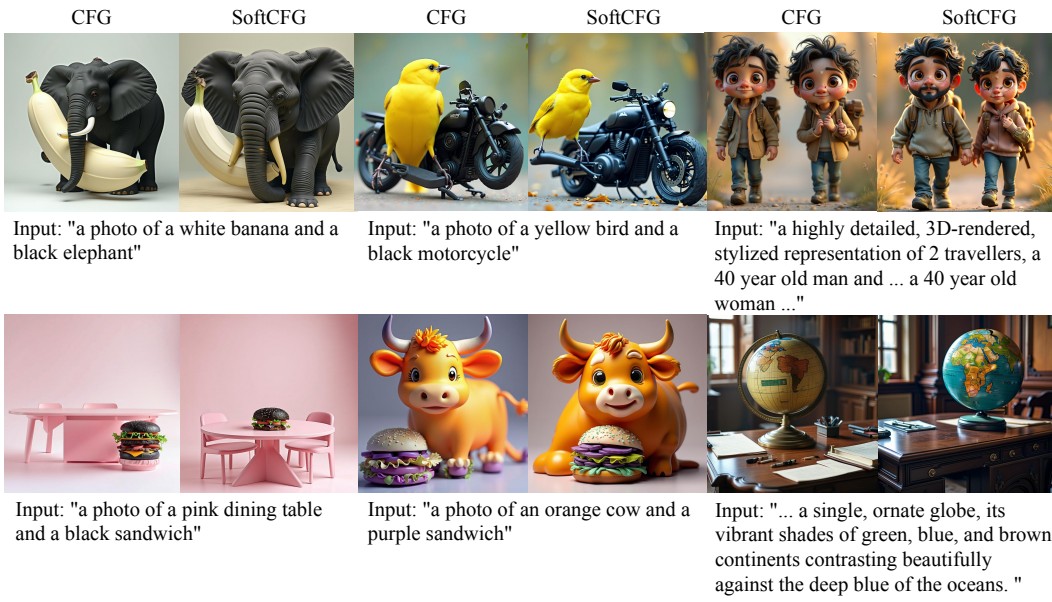

Figure 2: Comparison of images generated by standard Classifier-Free Guidance (CFG) and our proposed SoftCFG on LuminaGPT-8B Xin et al. (2025). Unlike CFG, which applies the same conditional offset regardless of generation history, SoftCFG adaptively incorporates uncertainty from the already generated content. As a result, SoftCFG effectively reduces unreasonable artifacts, such as motorcycles collapsing into tangled shapes, extra trunks emerging from nowhere, or redundant hands in humans. This demonstrates that by aligning guidance with generated content, SoftCFG yields more coherent and visually plausible generations.

Despite these advances, the quality of AR-based image generation is still heavily influenced by the inference process. Recent work (Sun et al., 2024) has introduced classifier-free guidance (CFG) (Ho & Salimans, 2022) into the AR setting, aiming to improve conditional generation by amplifying the difference between conditional and unconditional predictions during inference. As illustrated in Fig. 1 (a), in autoregressive models, CFG is typically applied by generating two parallel predictions at each generation step: one conditioned on the target class (by including a class token or embedding at the start of the input sequence), and one unconditional version where the first class token is replaced with an empty token. This technique, originally developed for diffusion models, has shown promising results (Yu et al., 2024a; Ren et al., 2025; Wu et al., 2025b) in steering AR generation towards class conditions without requiring auxiliary classifiers.

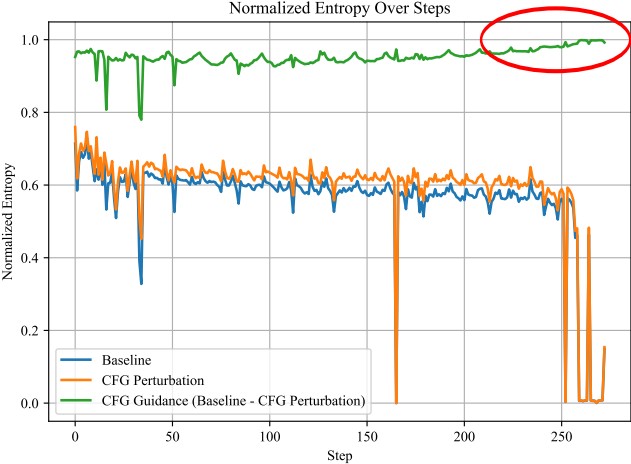

Figure 3: Diminishing effect of classifier-free guidance (CFG) in AR model Alitok-XL Wu et al. (2025b). We plot the normalized entropy over generation steps. As generation progresses, the difference (*i.e.*, the guidance signal, green line in the plot) between baseline (blue line) and CFG perturbation (orange line) entropy quickly vanishes. Here, a normalized entropy close to 1 indicates that guidance no longer provides informative guidance, please refer to Appendix C for more details of normalized entropy.

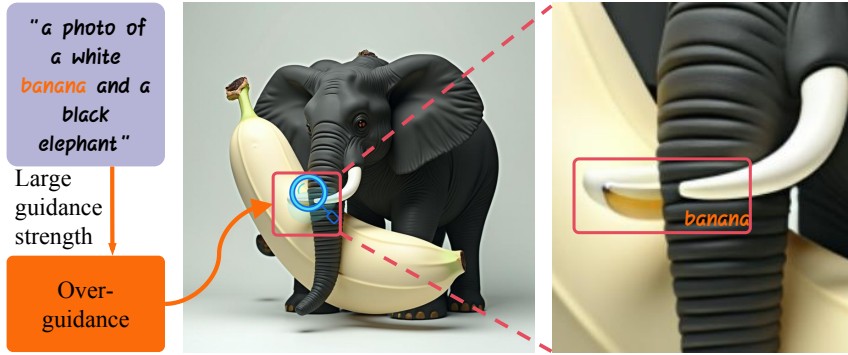

Figure 4: Illustration of the over-guidance phenomenon. When applying a large guidance strength, the model over-emphasizes certain words in the prompt (*e.g.*, *"banana"*), leading to distorted generations. In this example, the model incorrectly maps the word "banana" to the elephant's tusk, highlighting how excessive guidance strength can harm semantic alignment.

However, applying CFG to AR models introduces two fundamental challenges: **First, the guidance signal can *diminish* over time**: in contrast to diffusion, where guidance is injected at every denoising step, AR models rely heavily on conditioning tokens at the beginning of the sequence. As decoding proceeds, these tokens drift away from the local context, causing the conditional–unconditional gap to shrink and eventually vanish, even within short sequences (*e.g.*, $16 \times 16$ grids), as shown in Fig. 3. Some recent methods (Tian et al., 2024; Yu et al., 2024a; Han et al., 2025) mitigate this by injecting conditional embeddings directly into the prediction head (*e.g.*, AdaLN (Perez et al., 2018)), ensuring conditional information remains accessible.

**Second, CFG can also suffer from *over-guidance*.** Since guidance depends solely on external conditions such as class labels or text prompts, increasing the guidance scale often enforces semantics too aggressively, conflicting with visual coherence. This leads to artifacts such as duplicated limbs, redundant handlebars, or spurious object parts, as illustrated in Fig. 4. Even methods that inject conditional embeddings at every step remain tied to external signals and thus cannot fully resolve this semantic–visual conflict.

Together, this duality mirrors gradient vanishing and explosion in neural network training, which are usually mitigated by normalization and regularization methods. Motivated by this analogy, we propose **SoftCFG**: an uncertainty-guided inference method that distributes guidance more stably across the generation sequence. *The key idea behind SoftCFG is* **to let every generated token contribute certainty-weighted guidance, ensuring the signal persists across steps while naturally regularizing conflicts between text guidance and context**, as illustrated in Fig. 1 (b). In this work, we instantiate the weighting by prediction confidence which can be aligned well with the semantics of generated content (as illustrated in Fig. 5), but the framework is general and can accommodate learned scorers or perceptual alignment measures. Similar to how CFG perturbs the class token to encourage alignment with the class condition, SoftCFG perturbs high-confidence tokens to encourage future tokens to align with the most semantically reliable content generated so far. Therefore, SoftCFG not only alleviates the fading guidance problem but also reconciles the conflict between semantic alignment and visual coherence, leading to more stable and plausible generations.

To realize this idea, we need a mechanism that allows generated tokens to directly influence future decoding steps. Since the value cache stores the representations of past tokens that are repeatedly attended to by subsequent predictions, it provides a natural handle for injecting token-wise guidance. We therefore compute each token's maximum predicted probability $p_{\max}$ as a measure of its confidence, and scale its value cache across all attention layers by a factor of $(1 - p_{\max})$ during inference, as illustrated in Figure 6(b). In this way, tokens with higher confidence receive stronger perturbations on their cached representations, which amplifies their downstream impact and distributes guidance signals throughout the sequence. To avoid degenerate behavior when many tokens accumulate large perturbations or when $p_{\max}$ approaches 1, we further introduce a **Step Normalization** procedure: at each step, we normalize the set of perturbation weights so that their sum remains constant (*e.g.*, 1), ensuring stable and balanced guidance throughout the generation process.

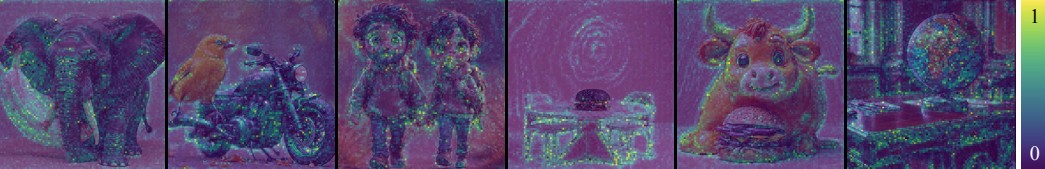

Figure 5: Heatmaps of token confidence overlaid on generated images by LuminamGPT2 (Xin et al., 2025). High-confidence regions align well with salient semantic structures (*e.g.*, object parts), while low-confidence regions occur in ambiguous backgrounds, supporting high-confidence tokens can be effective guidance signals.

SoftCFG requires **no additional training**, introduces **no architectural change**, and adds **negligible computational overhead**, making it fully compatible with existing AR models and CFG setups. By leveraging uncertainty-weighted perturbations and step normalization, it improves both the stability and effectiveness of guidance during inference. Experiments on ImageNet show that SoftCFG significantly improves generation quality, reducing the FID from **1.37** to **1.27** over a SOTA AR baseline (Wu et al., 2025b), setting a new state of the art for autoregressive models on $256 \times 256$ Deng et al. (2009) benchmark.

We conclude our contributions as follows:

- We demonstrate the necessity of leveraging generated visual content as guidance for subsequent token prediction, extending the notion of guidance beyond class or text tokens.

- We propose SoftCFG, a general framework that introduces token-wise soft weighting to integrate such guidance. In this work, we instantiate the weighting by token confidence, while other scoring functions or learned modules can be naturally incorporated.

- Through comprehensive experiments on class-conditional benchmarks (*e.g.*, ImageNet $256 \times 256$), we show that SoftCFG achieves state-of-the-art FID among autoregressive models, consistently improving quality and alignment with negligible overhead.

## 2 SOFT CLASSIFIER-FREE GUIDANCE

### 2.1 PRELIMINARY: CLASSIFIER-FREE GUIDANCE IN AR MODELS

Classifier-Free Guidance (CFG) is a widely used technique to enhance controllability in generative models by interpolating predictions from conditional and unconditional branches. At each decoding step $t$, an autoregressive model maintains key/value caches $(\mathbf{K}_{<t}, \mathbf{V}_{<t})$ summarizing previously generated tokens $x_{<t}$. The branch logits are

$$\mathbf{z}_t^{\text{cond}} = f_\theta(\mathbf{K}_{<t}^{\text{cond}}, \mathbf{V}_{<t}^{\text{cond}}, x_{t-1}, c), \quad \mathbf{z}_t^{\text{uncond}} = f_\theta(\mathbf{K}_{<t}^{\text{uncond}}, \mathbf{V}_{<t}^{\text{uncond}}, x_{t-1}, \emptyset), \tag{1}$$

and the guided logits are formed as

$$\mathbf{z}_t^{\text{CFG}} = \mathbf{z}_t^{\text{uncond}} + \gamma \left( \mathbf{z}_t^{\text{cond}} - \mathbf{z}_t^{\text{uncond}} \right), \quad p_\theta^{\text{CFG}}(\cdot \mid x_{<t}, c) = \text{softmax}(\mathbf{z}_t^{\text{CFG}}), \tag{2}$$

where $\gamma > 0$ is the guidance scale. Here, we use $\Delta_t = \mathbf{z}_t^{\text{cond}} - \mathbf{z}_t^{\text{uncond}}$ to denote the step-wise guidance offset, which is only decided by input text condition.

Standard CFG applies a *hard offset* $\gamma \Delta_t$ at every step. While this improves conditional alignment, it can also *over-amplify* the conditional signal, especially when $\|\Delta_t\|$ or $\gamma$ is large. This phenomenon, which we term *over-guidance* as illustrated in Fig. 4, often manifests as structural artifacts (*e.g.*, duplicated parts or implausible object completions). Moreover, in AR models, once $x_t$ is sampled, this offset does not propagate explicitly to future steps, so the effect of $\Delta_t$ may *decay too quickly* across long sequences, as shown in Fig. 3. These two limitations motivate the development of a softer, regularized alternative to CFG.

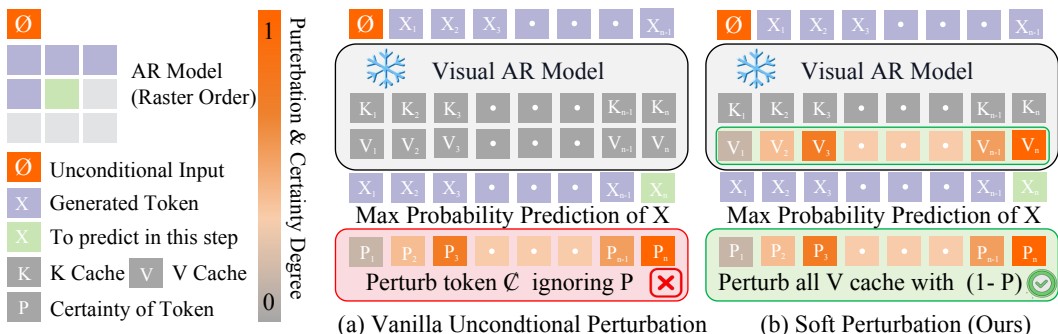

Figure 6: Two perturbation strategies for Visual AR models. (a) Unconditional Perturbation modifies the first class conditional token regardless of the certainty score. (b) Uncertainty-guided Perturbation applies softer, weighted changes to all $\mathbf{V}$ cache entries, with strength $(1 - \mathbf{P})$, offering stronger perturbation to high-confidence tokens.

## 2.2 SOFTCFG: UNCERTAINTY-GUIDED PERTURBATION

At a high level, our goal is to address the over-guidance and guidance diminishing issues identified in Sec. 2.1, and make guidance both *context-aware* and *stable* over long horizons. Instead of relying solely on external conditions, we incorporate already generated content and regularize guidance through the model's context. Concretely, SoftCFG acts as a context-aware regularizer on the *unconditional* branch: it softly adjusts internal states so that reliable tokens contribute more and uncertain ones less. This redistribution embeds guidance into the model's memory, enabling it to persist across future decoding steps—without retraining or architectural changes.

Specifically, for each previously generated token $i < t$, we compute its predictive confidence using the maximum probability from the conditional distribution:

$$w_i = 1 - p_{\max}(x_i), \qquad p_{\max}(x_i) = \max_v \; p_\theta^{\text{cond}}(\cdot \mid x_{<i}, c). \tag{3}$$

Tokens with higher confidence (low $w_i$) are considered more reliable and thus receive stronger perturbations. We also test different confidence indicators in Sec. 3.4. We scale their unconditional value vectors as:

$$\tilde{\mathbf{v}}_i^{\text{uncond, pertcontext}} = w_i \, \mathbf{v}_i^{\text{uncond}}, \tag{4}$$

where $\tilde{\mathbf{v}}_i^{\text{uncond, pertcontext}}$ denotes this value cache has no conditional information and its context information is also perturbed. Then we can produce context-perturbed unconditional logits as follows:

$$\tilde{\mathbf{z}}_t^{\text{uncond, pertcontext}} = f_\theta(\mathbf{K}_{<t}^{\text{uncond}}, \tilde{\mathbf{V}}_{<t}^{\text{uncond, pertcontext}}, x_{t-1}, \emptyset). \tag{5}$$

SoftCFG then combines the conditional branch with the perturbed unconditional branch:

$$\mathbf{z}_t^{\text{SoftCFG}} = \mathbf{z}_t^{\text{cond}} + \gamma \, (\mathbf{z}_t^{\text{cond}} - \tilde{\mathbf{z}}_t^{\text{uncond, pertcontext}}). \tag{6}$$

In this way, SoftCFG weakens the unconditional context based on token confidence: reliable tokens are downscaled more, while uncertain ones are preserved. This adaptive adjustment redefines the guidance itself: instead of relying solely on the external condition, SoftCFG incorporates normalized context from already generated tokens, allowing the guidance signal to be maintained and propagated consistently across future steps.

Equivalently, SoftCFG can be written as a regularized variant of CFG in Eq. 2:

$$\mathbf{z}_t^{\text{SoftCFG}} = \mathbf{z}_t^{\text{CFG}} + \gamma \, \Delta_t^{\text{context}}, \qquad \Delta_t^{\text{context}} = \mathbf{z}_t^{\text{uncond}} - \tilde{\mathbf{z}}_t^{\text{uncond, pertcontext}}. \tag{7}$$

The correction $\Delta_t^{\text{context}}$ is induced by selectively weakening the unconditional value cache, and hence is bounded by the perturbation weights $\{w_i\}$.

Geometrically, the additional term $\Delta_t^{\text{context}}$ acts as a *context-aware regularizer*. When $\Delta_t^{\text{context}}$ is aligned with the original guidance offset $\Delta_t = \mathbf{z}_t^{\text{cond}} - \mathbf{z}_t^{\text{uncond}}$, SoftCFG amplifies the conditional signal; when misaligned, it shrinks it. This mirrors the role of classical $\ell_2$ regularization, which constrains parameter updates by pulling them back toward the origin. Here, instead of parameters, we regularize the *contextual representations*, ensuring that the guidance signal remains bounded and smoothly propagated through the sequence.

---

**Algorithm 1** SoftCFG sampling with Step Normalization

---

1: Initialize $x_0 \leftarrow \langle bos \rangle$; empty caches $\{\mathbf{K}^{\text{cond}}, \mathbf{V}^{\text{cond}}, \mathbf{K}^{\text{uncond}}, \mathbf{V}^{\text{uncond}}, \mathbf{P}_{\text{max}}\}$
2: **for** $t = 1$ to $T$ **do**
3:     $\mathbf{z}_t^{\text{cond}} \leftarrow f_\theta(\mathbf{K}_{<t}^{\text{cond}}, \mathbf{V}_{<t}^{\text{cond}}, x_{t-1}, c)$
4:     **for** $i = 1, \ldots, t-1$: $w_i \leftarrow 1 - p_{\text{max}}(x_i)$         ▷ get the confidence of all generated token
5:     **for** $i = 1, \ldots, t-1$: $\hat{w}_i \leftarrow 1 - \dfrac{1 - w_i}{\sum_{j=1}^{t-1}(1 - w_j) + \varepsilon}$         ▷ step normalization
6:     $\tilde{\mathbf{V}}_{<t}^{\text{uncond}} \leftarrow \mathbf{V}_{<t}^{\text{uncond}}$;    **for** $i = 1, \ldots, t-1$: $\tilde{\mathbf{v}}_i^{\text{uncond}} \leftarrow \hat{w}_i \cdot \mathbf{v}_i^{\text{uncond}}$     ▷ apply soft perturbation
7:     $\tilde{\mathbf{z}}_t^{\text{uncond,pertcontext}} \leftarrow f_\theta(\mathbf{K}_{<t}^{\text{uncond}}, \tilde{\mathbf{V}}_{<t}^{\text{uncond}}, x_{t-1}, \emptyset)$
8:     $\mathbf{z}_t^{\text{SoftCFG}} \leftarrow (1 + \gamma)\mathbf{z}_t^{\text{cond}} - \gamma\,\tilde{\mathbf{z}}_t^{\text{uncond,pertcontext}}$       ▷ SoftCFG with context-regularization
9:     $x_t \sim \text{Sample}(\text{softmax}(\mathbf{z}_t^{\text{SoftCFG}}))$
10:    Update *original* caches with $x_t$ to obtain $\{\mathbf{K}_{\leq t}^{\text{cond}}, \mathbf{V}_{\leq t}^{\text{cond}}, \mathbf{K}_{\leq t}^{\text{uncond}}, \mathbf{V}_{\leq t}^{\text{uncond}}, \mathbf{P}_{\text{max}}\}$;
11: **end for**

---

## 2.3 STEP NORMALIZATION

However, we observed an explosion of $\Delta_t^{\text{context}}$ in Fig. 7, where the normalized entropy of the guidance increases rapidly, indicating that unbounded SoftCFG may cause the guidance to explode. We attribute this to the cumulative growth of perturbations in the unconditional branch.

Formally, the deviation of SoftCFG from vanilla CFG, defined by the context-aware regularizer $\Delta_t^{\text{context}}$ in Eq. 7, is bounded by:

$$\|\Delta_t^{\text{context}}\| \leq L_t \cdot \sum_{i<t}(1 - w_i)\|\mathbf{v}_i^{\text{uncond}}\|, \tag{8}$$

where $L_t$ is the Lipschitz constant of $f_\theta$ with respect to the value cache. Please refer to Appendix D for more details. If $\sum_{i=1}^{t-1}(1 - w_i)$ were bounded, the deviation would be at most $O(\gamma)$ within a controlled trust region.

However, as the sequence length $t$ increases, the cumulative deviation $\sum_{i=1}^{t-1}(1 - w_i)$ can grow significantly, especially when many tokens have high confidence ($w_i \to 0, 1 - w_i \to 1$), potentially causing $\Delta_t^{\text{context}}$ to become excessively large and leading SoftCFG to deviate far from vanilla CFG. This also explains the degeneration observed in practice: *the unconditional branch gradually loses all contextual information, causing instability in long-horizon generation.* Formally, if $w_i \to 0$ for high-confidence tokens, the contribution of $\mathbf{V}_{<t}^{\text{uncond}}$ is suppressed and the unconditional branch degenerates: $\mathbb{E}[\tilde{\mathbf{z}}_t^{\text{uncond,pertcontext}}] \to 0$.

To mitigate this, we introduce **Step Normalization**, which renormalizes the perturbation weights at every step:

$$\hat{w}_i = 1 - \frac{1 - w_i}{\sum_{j=1}^{t-1}(1 - w_j)} \quad \text{such that} \quad \sum_{i=1}^{t-1}(1 - \hat{w}_i) = 1. \tag{9}$$

**Proposition 1** (Bounded Deviation of Step-Normalized SoftCFG). *Let $f_\theta$ be $L_t$-Lipschitz with respect to its value-cache input at step $t$. Then the deviation of SoftCFG from vanilla CFG is bounded as*

$$\|\Delta_t^{context}\| = \|\tilde{\mathbf{z}}_t^{uncond,pert} - \mathbf{z}_t^{uncond}\| \leq L_t \cdot \max_{i<t}\|\mathbf{v}_i^{uncond}\|, \tag{10}$$

*where $\tilde{\mathbf{z}}_t^{uncond,pert}$ denotes the unconditional logits under step-normalized perturbation.*

Without normalization, the cumulative perturbation of past tokens grows proportionally with sequence length. Step normalization rescales the perturbation weights $\{1 - w_i\}$ so that $\sum_{i=1}^{t-1}(1 - \hat{w}_i) = 1$ at each step, effectively allocating a unit perturbation budget over the context. Therefore the perturbation magnitude at step $t$ is controlled by at most one token's contribution, leading to the bound $\|\Delta_t^{\text{context}}\| \leq L_t \cdot \max_{i<t}\|\mathbf{v}_i^{\text{uncond}}\|$.

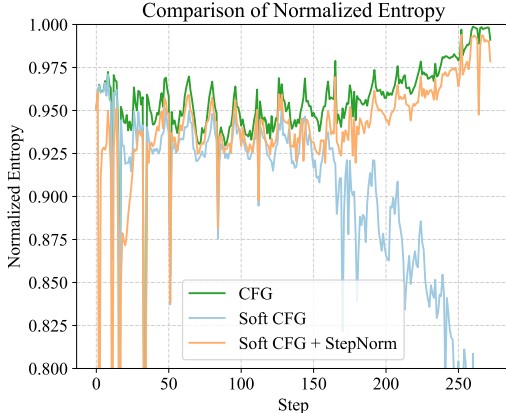 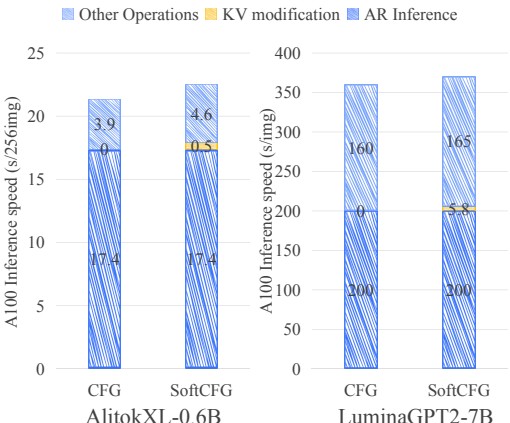

Figure 7: Comparison of normalized entropy across different sampling strategies. Step normalization stabilizes SoftCFG by controlling cumulative perturbations of the unconditional branch.

Figure 8: Inference speed comparison of CFG and SoftCFG (w Step Norm) on AliTok-0.6B ($256 \times 256$) and LuminaGPT2-7B ($756 \times 756$). SoftCFG adds negligible overhead.

## 3  EXPERIMENTS

### 3.1  SETUP

**Dataset and Benchmark.**  For *class-conditional* generation, we evaluate on ImageNet (Deng et al., 2009) with the standard $256 \times 256$ class-conditional generation protocol. For *text-to-image* generation, we evaluate on two widely used benchmarks, *i.e.*, GenEval benchmark (Ghosh et al., 2023) and DPG-Bench (Hu et al., 2024).

**Models.**  We evaluate SoftCFG on state-of-the-art (SOTA) autoregressive (AR) generators to ensure that improvements are not due to weak baselines. For *class-conditional* generation, we use the strongest published AR model on ImageNet $256 \times 256$ (Deng et al., 2009), **AliTok** (Wu et al., 2025b). For *text-to-image* generation, we use **LuminaGPT2** Xin et al. (2025), a recently released large-scale AR model that achieves the best performance on both GenEval Ghosh et al. (2023) and DPG-Bench (Hu et al., 2024). All models adopt a VQ-style discrete tokenizer and support classifier-free guidance (CFG) at inference. Our method, *SoftCFG*, is applied only at inference, without retraining or modifying the architecture.

**Metrics.**  For *class-conditional* generation task, we generate $50$k samples on the validation label set and report Fréchet Inception Distance (FID; lower is better). For completeness, we also track Inception Score (IS), precision/recall for generative models.

**Hyperparameter settings.**  We follow the baseline sampling setup and keep all decoding hyperparameters identical to CFG (temperature, guidance scale $\gamma$, *etc.*). SoftCFG uses token-wise uncertainty weights $w_i = 1 - p_{\max}(x_i)$ measured on the *unconditional* branch at the time token $x_i$ is generated, and perturbs only the unconditional value cache as defined in Sec. 2.2. Guidance scale $\gamma$ matches the CFG baseline unless stated otherwise.

### 3.2  IMPLICATIONS AND TIME COMPLEXITY ANALYSIS

We summarize the inference procedure of SoftCFG with step normalization in Alg. 1. At each decoding step, the conditional logits are computed as in vanilla CFG, while the unconditional branch is temporarily perturbed by rescaling its value-cache entries according to token-wise confidence. Step normalization ensures that the perturbation weights are re-normalized at every step, allocating a fixed perturbation budget across the context. **Importantly, the perturbation is applied by an in-place scalar rescaling of the stored unconditional V-cache, which is negligible compared to attention/MLP compute;** As illustrated in Fig. 8, the SoftCFG sampling time matches CFG.

Table 1: ImageNet-1K (Deng et al., 2009) $256 \times 256$ generation results evaluated with ADM (Dhariwal & Nichol, 2021). * indicates results from our own implementation.

| Type | Generator | Venue | #Params | FID↓ | IS↑ | sFID↓ | Pre.↑ | Rec.↑ |
|---|---|---|---|---|---|---|---|---|
| Diff. | LDM-8 Rombach et al. (2022) | CVPR 22 | 258M | 7.76 | 209.5 | - | 0.84 | 0.35 |
| | LDM-4 Rombach et al. (2022) | CVPR 22 | 400M | 3.60 | 247.7 | - | **0.87** | 0.48 |
| | UViT-L/2 Bao et al. (2023) | CVPR 23 | 287M | 3.40 | 219.9 | - | 0.83 | 0.52 |
| | UViT-H/2 Bao et al. (2023) | CVPR 23 | 501M | 2.29 | 263.9 | - | 0.82 | 0.57 |
| | DiT-L/2 Peebles & Xie (2023) | ICCV 23 | 458M | 5.02 | 167.2 | - | 0.75 | 0.57 |
| | DiT-XL/2 Peebles & Xie (2023) | ICCV 23 | 675M | 2.27 | 278.2 | 4.60 | 0.83 | 0.57 |
| | SiT-XL Ma et al. (2024) | ECCV 24 | 675M | 2.06 | 270.3 | 4.50 | 0.82 | 0.59 |
| | DiMR-XL/2R Liu et al. (2024) | NeuIPS 24 | 505M | 1.70 | 289.0 | - | 0.79 | 0.63 |
| | MDTV2-XL/2 Gao et al. (2023) | ICCV 23 | 676M | 1.58 | 314.7 | 4.52 | 0.79 | 0.65 |
| | REPA Yu et al. (2025) | ICLR 25 | 675M | 1.42 | 305.7 | 4.70 | 0.80 | 0.65 |
| | REPA-E Leng et al. (2025) | ICCV 25 | 675M | **1.26** | **314.9** | 4.11 | 0.79 | **0.66** |
| Mask. | MaskGIT Chang et al. (2022) | CVPR 22 | 177M | 6.18 | 182.1 | - | 0.80 | 0.51 |
| | TiTok-S-128 Yu et al. (2024b) | NeuIPS 24 | 287M | 1.97 | 281.8 | - | - | - |
| | MAGVIT-v2 Yu et al. (2023) | CVPR 23 | 307M | 1.78 | 319.4 | - | - | - |
| | MaskBit Weber et al. (2024) | Arxiv 24 | 305M | **1.52** | **328.6** | - | - | - |
| VAR | VAR-d30 Tian et al. (2024) | NeuIPS 24 | 2.0B | 1.92 | 323.1 | - | 0.82 | 0.59 |
| | VAR-d30-re Tian et al. (2024) | NeuIPS 24 | 2.0B | **1.73** | 325.0 | - | 0.82 | 0.60 |
| MAR | MAR-B Li et al. (2024) | NeuIPS 24 | 208M | 2.31 | 281.7 | - | 0.82 | 0.57 |
| | MAR-L Li et al. (2024) | NeuIPS 24 | 479M | 1.78 | 296.0 | - | 0.81 | 0.61 |
| | MAR-H Li et al. (2024) | NeuIPS 24 | 943M | 1.55 | 303.7 | - | 0.81 | 0.62 |
| FlowAR | FlowAR-S Ren et al. (2025) | ICML 25 | 170M | 3.61 | 234.1 | - | 0.83 | 0.50 |
| | FlowAR-H Ren et al. (2025) | ICML 25 | 1.9B | 1.65 | 296.5 | - | 0.83 | 0.60 |
| AR | GPT2 Esser et al. (2021) | CVPR 21 | 1.4B | 15.78 | 74.3 | - | - | - |
| | GPT2-re Esser et al. (2021) | CVPR 21 | 1.4B | 5.20 | 280.3 | - | - | - |
| | VIM-L Yu et al. (2022) | ICLR 22 | 1.7B | 4.17 | 175.1 | - | - | - |
| | VIM-L-re Yu et al. (2022) | ICLR 22 | 1.7B | 3.04 | 227.4 | - | - | - |
| | Open-MAGVIT2-B Luo et al. (2024) | Arxiv 24 | 343M | 3.08 | 258.3 | - | **0.85** | 0.51 |
| | Open-MAGVIT2-XL Luo et al. (2024) | Arxiv 24 | 1.5B | 2.03 | 286.0 | - | 0.84 | 0.54 |
| | LlamaGen-L Sun et al. (2024) | Arxiv 24 | 343M | 3.07 | 256.1 | - | 0.83 | 0.52 |
| | LlamaGen-3B Sun et al. (2024) | Arxiv 24 | 3.1B | 2.18 | 263.3 | - | 0.81 | 0.58 |
| | RandAR-L Pang et al. (2025) | CVPR 25 | 343M | 2.55 | 288.8 | - | 0.81 | 0.58 |
| | RandAR-XXL Pang et al. (2025) | CVPR 25 | 1.4B | 2.15 | 322.0 | - | 0.79 | 0.62 |
| | RAR-B (Yu et al., 2024a) | Arxiv 24 | 261M | 1.95 | 290.5 | - | 0.82 | 0.58 |
| | RAR-XL (Yu et al., 2024a) | Arxiv 24 | 955M | 1.50 | 306.9 | - | 0.80 | 0.62 |
| | Alitok-B (Wu et al., 2025b) | Arxiv25 | 177M | 1.50 | 305.9 | - | 0.78 | 0.64 |
| | Alitok-L (Wu et al., 2025b) | Arxiv25 | 318M | 1.42 | **326.6** | - | 0.78 | 0.65 |
| | Alitok*-XL (Wu et al., 2025b) | Arxiv25 | 662M | 1.35 | 317.1 | 6.96 | 0.79 | 0.64 |
| | Alitok-B*+SoftCFG (ours) | - | 177M | 1.40 | 271.0 | **5.95** | 0.78 | 0.66 |
| | Alitok-L*+SoftCFG (ours) | - | 318M | 1.39 | 272.3 | 6.00 | 0.78 | **0.66** |
| | Alitok-XL*+SoftCFG (ours) | - | 662M | **1.27** | 302.4 | 6.76 | 0.78 | 0.65 |

## 3.3 State-of-the-art Comparisons.

**Conditional Image Generation.** Table 1 summarizes results on ImageNet-1K $256 \times 256$. Diffusion and masking models achieved strong IS and better FID. Recent AR models (*e.g.*, LlamaGen Sun et al. (2024), RandAR Pang et al. (2025)) reduce this gap. Our AliTok+SoftCFG attains an FID of 1.27, narrowing the gap between AR and diffusion models, while maintaining competitive IS and improved recall. With SoftCFG, AR demonstrates stronger potential to replace diffusion in future architectural designs.

## 3.4 Ablations

**Effect of SoftCFG** As shown in Table 2, replacing vanilla CFG with SoftCFG reduces FID (1.32 vs. 1.37), indicating more effective guidance. We also show the visualization comparison on LuminamGPT2 (Xin et al., 2025) in Fig. 1. However, as discussed in Sec. 2.3, directly applying Soft-CFG could make the unconditional branch lose too much contextual information, therefore SoftCFG slightly destabilizes IS due to amplified step-wise perturbations.

**Effect of Step Normalization** Adding step normalization on top of SoftCFG mitigates the above issue, yielding both stable IS and improved sFID (7.16). This confirms that step normalization complements SoftCFG by regularizing its perturbation strength over time. Figure 7 illustrates the normalized entropy on AliTok-XL with $\gamma = 13$, $k = 14$, where the confidence score is defined as the conditional probability after guidance.

Table 2: Ablation study on ImageNet $256\times256$ class-conditional generation. We use the optimal $\gamma = 13$ (in Eq. 2) and $k = 1.4$ (in Sec. 3.4) for CFG on Alitok-XL Wu et al. (2025b). We report FID, IS and sFID. SoftCFG improves FID over vanilla CFG, and step normalization further stabilizes IS and sFID. Opt. indicates that we jointly tune $(\gamma, k)$ via a small grid. We bold the top-2 values for each metric.

| Method | FID ↓ | IS ↑ | sFID ↓ |
|---|---|---|---|
| Baseline | 1.76 | 221.2 | **5.55** |
| + CFG + Opt. | 1.35 (-0.41) | **317.1** (+95.9) | 6.96 (+1.41) |
| + SoftCFG | **1.32** (-0.44) | 288.1 (+66.9) | 7.62 (+2.07) |
| + SoftCFG + StepNorm | **1.32** (-0.44) | 302.0 (+80.8) | 7.16 (+1.61) |
| + SoftCFG + StepNorm + Opt. | **1.27** (-0.49) | **302.4** (+81.2) | **6.70** (+1.15) |

Table 3: Effect of perturbing different cache components in AliTok-XL on ImageNet-256. SoftCFG with perturbations applied only to the value cache achieves the best FID.

| SoftCFG | FID ↓ | IS ↑ | sFID ↓ | Precision ↑ | Recall ↑ |
|---|---|---|---|---|---|
| Perturb **K and V** | 1.31 | 287.6 | 6.94 | 0.79 | **0.65** |
| Perturb **K** only | 1.39 | **302.5** | 7.66 | **0.80** | 0.64 |
| Perturb **V** only | **1.27** | 302.4 | **6.70** | 0.79 | **0.65** |

**Effect of V perturbation**  We ablate which part of the attention cache SoftCFG perturbs. Since queries of generated tokens are not needed in attention calculation, we only compare perturbing **V**, **K**, or both **K** and **V** on AliTok-XL (ImageNet-256, 50k samples). As shown in Table 3, perturbing **V** only gives the best FID/sFID and highest recall while keeping IS and precision competitive, whereas perturbing **K** (with or without **V**) slightly improves IS/precision but consistently worsens FID, sFID, and recall, suggesting that modifying **K** disrupts attention routing. This empirically supports our design choice of perturbing **V** as the most robust and practical option for AR decoding.

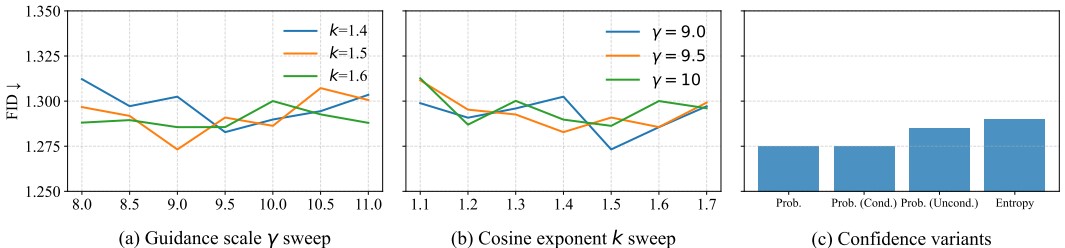

(a) Guidance scale $\gamma$ sweep  (b) Cosine exponent $k$ sweep  (c) Confidence variants

Figure 9: Adaptive hyperparameter ablations of SoftCFG. (a) Guidance scale $\gamma$ sweep across different cosine exponents $k$. (b) Cosine exponent $k$ sweep under fixed guidance scales $\gamma$. (c) Comparison of different confidence score definitions. Results show that SoftCFG is robust to hyperparameter choices and that the max probability of conditional branch yields the best trade-off.

**Hyperparameter Varying**  We analyze the sensitivity of SoftCFG to (i) the guidance scale $\gamma$ (in Eq. 2 and Eq. 6) and (ii) the power parameter $k$ in cosine scheduling introduced by Yu et al. (2024a). Here, a cosine schedule for the guidance scale is: $\gamma_t = (\gamma - 1) \cdot \frac{1}{2}(1 - \cos((t/T)^k \pi))$, where $t$ is the current step and $k$ controls how guidance is distributed: larger $k$ shifts it to later steps, smaller $k$ to earlier steps. As shown in Fig. 9 (a) and Fig. 9 (b), SoftCFG consistently improves FID across a wide range of $\gamma$, while vanilla CFG quickly deteriorates for large scales. For cosine scheduling, moderate powers ($k = 1.5$ or $1.6$) yield the best trade-off, whereas extreme values hurt performance. These results highlight that **tuning is still required, and the effective ranges of $\gamma$ and $k$ are broader and the optimal values are typically smaller than those of vanilla CFG.** This is because SoftCFG generally produces stronger guidance magnitudes than vanilla CFG. In addition, Fig. 9 (c) compares different definitions of the confidence score. We find that using the max probability of the conditional branch yields the most stable improvements, further showing that SoftCFG is robust to confidence variations.

Table 4: SoftCFG on text-to-image generation with SimpleAR. We report overall score and fine-grained compositional metrics.

| Method | Overall | % Correct Images | % Correct Prompts | Single Obj. | Two Obj. | Position | Colors | Color Attr. | Counting |
|---|---|---|---|---|---|---|---|---|---|
| **SimpleAR** | **0.61744** | 61.12% | 80.65% | 100.00% | 89.39% | **30.25%** | 81.38% | 36.00% | **33.44%** |
| **+ SoftCFG** | 0.61266 | 60.80% | 80.29% | 99.69% | **91.41%** | 26.00% | **82.18%** | **40.50%** | 27.81% |

Figure 10: Qualitative comparison between CFG and SoftCFG on text-to-image generation. Soft-CFG yields cleaner, more coherent foregrounds with fewer artifacts, but occasionally misplaces objects *w.r.t.* spatial keywords. The DPG-Bench Hu et al. (2024) evaluator relies on strict rule-based text–image alignment and largely ignores image quality.

**Text-to-image Generation** Table 4 shows that adding SoftCFG to SimpleAR slightly lowers the overall DPG-Bench Hu et al. (2024) score (0.617→0.613), but improves several compositional aspects: two-object prompts (89.39%→91.41%), color consistency (81.38%→82.18%), and color-attribute correctness (36.00%→40.50%). As shown in Fig. 10, position remain challenging and show small regressions, suggesting that **current text-to-image evaluation focuses on strict rule-based text-image alignment instead of image quality and coherence**. Overall, SoftCFG enhances image quality in T2I AR models, while leaving room for future work on text-image alignment.

## 4    CONCLUSION

In this work, we introduced SoftCFG, a lightweight modification to classifier-free guidance for autoregressive generation. Our key insight is that, similar to class tokens, previously generated visual content can also provide guidance to subsequent tokens. To fully exploit this effect, SoftCFG applies a soft weighting mechanism that adaptively modulates the influence of past tokens. While in this work we instantiated the weights with token-level confidence, our framework naturally accommodates more sophisticated scoring functions, including alternative scoring mechanisms, such as learned discriminators or perceptual evaluators. Extensive experiments on class-conditional generation demonstrate that SoftCFG consistently improves fidelity and alignment without retraining or increasing inference cost. We believe this simple yet general principle opens new directions for integrating fine-grained guidance signals into autoregressive generation.

ACKNOWLEDGMENTS

We acknowledge funding from the Flemish Government (AI Research Program) and the Research Foundation - Flanders (FWO) through project number **G0G2921N**. This project was also supported by Sigrid Juselius Foundation and the Finnish Research Council (**Profi6 336449 funding program**), the strategic funding of the University of Oulu. We acknowledge EuroHPC JU for awarding the project ID **EHPC-AIF-2025SC02-103** access to **MareNostrum5 ACC hosted by BSC at Barcelona, Spain.**

ETHICS&REPRODUCIBILITY STATEMENT

This work relies only on public datasets, and while generative models may be misused, our method does not increase such risks. All key settings are reported, and we have released the core code for SoftCFG.

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

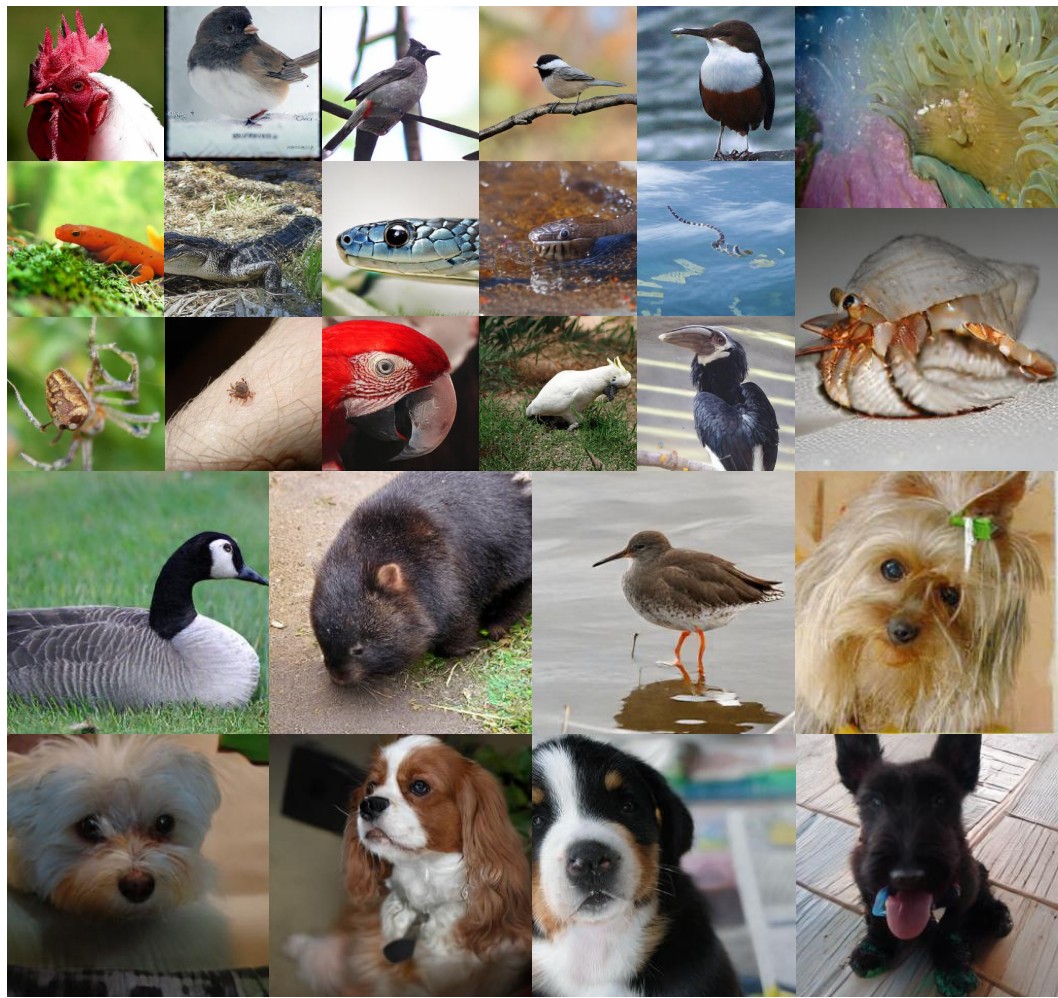

Figure 11: Samples generated by AliTok-XL with SoftCFG and Step Normalization on ImageNet $256 \times 256$. The results demonstrate high visual fidelity and semantic consistency across diverse classes.

## A  APPENDIX: VISUALIZATION

We also provide qualitative samples generated by AliTok-XL with SoftCFG and Step Normalization on ImageNet $256 \times 256$ (Fig. 11). These results illustrate the visual diversity and semantic fidelity that can be achieved with our method, complementing the quantitative evaluations in Sec. 3.4.

## B  APPENDIX: RELATED WORK

### B.1  VISUAL AR MODELS

Recent advances have significantly improved AR-based generation by enhancing both the visual tokenizer (Yu et al., 2024b; Wu et al., 2025b; Li et al., 2025b; Qu et al., 2025; Shi et al., 2025) and the generation paradigm (Tian et al., 2024; Pang et al., 2025; Yu et al., 2024a; Xu et al., 2025; Wang et al., 2025), leading to sharper spatial alignment and more coherent semantics. As a result, AR models (Wu et al., 2025b; Xu et al., 2025) have recently demonstrated generation quality on par with state-of-the-art diffusion-based and flow-based models (Yu et al., 2025; Leng et al., 2025) on standard benchmarks such as ImageNet (Deng et al., 2009).

## B.2 CLASSIFIER-FREE GUIDANCE IN GENERATIVE MODELS

Classifier-Free Guidance (CFG) was first introduced in diffusion models to improve conditional generation performance without relying on external classifiers. It interpolates between conditional and unconditional predictions during sampling, trading off between sample fidelity and diversity (Ho & Salimans, 2022). Since then, CFG has become a fundamental technique in generative modeling.

A growing body of research explores ways to refine or extend CFG, particularly by injecting perturbations or structural modifications to guidance: Self-Attention Guidance (Hong et al., 2023) proposes using attention-based modifications to guide generation in diffusion models, improving sample quality without needing new objectives. This work highlights the importance of internal structure manipulation as an alternative to direct logits-based guidance. Perturbed-Attention Guidance (Ahn et al., 2024) also introduces perturbed-attention into diffusion sampling to correct misaligned attention signals. Smoothed Energy Guidance (Hong, 2024) further addresses unstable energy landscapes in CFG by smoothing the attention curvature, resulting in more consistent guidance effects across different prompts and steps. Guiding a Diffusion Model with a Bad Version of Itself (Karras et al., 2024) shows that strong generation quality can be achieved by replacing the unconditional branch in CFG with a smaller or less-trained model.

Several recent works (Shen et al., 2024; Rajabi et al., 2025; Li et al., 2025a) also focus on improving CFG via dynamic and token-level perturbations: Semantic-aware CFG (Shen et al., 2024), which adjusts guidance based on spatial semantic regions to mitigate inconsistency. Token Perturbation Guidance (Rajabi et al., 2025), which locally perturbs input tokens or hidden states to encourage more robust generation paths. Adaptive CFG (A-CFG) (Li et al., 2025a), which applies dynamic low-confidence masking to the unconditional branch based on step-wise uncertainty.

While these approaches offer significant improvements for diffusion and masked generative models, they are not directly transferable to AR generation, where conditioning signals degrade rapidly due to the sequential nature of decoding. One of the few works to examine CFG in AR setting is Condition Contrastive Alignment (CCA) (Chen et al., 2024), which enables guidance-free sampling by fine-tuning the model to align conditional and unconditional outputs. However, it still underperforms standard CFG in most cases and requires additional training. In contrast, our method introduces Soft-CFG, a lightweight uncertainty-aware guidance mechanism for AR models, designed to stabilize the guidance signal across the entire generation sequence without modifying training, architecture, or inference efficiency. Beyond improving AR inference quality, SoftCFG provides a more stable and informative form of conditional guidance, which can potentially benefit future alignment-based or distillation-based approaches such as CCA and Model-guidance (Tang et al., 2025).

## C APPENDIX: NORMALIZED ENTROPY COMPUTATION

Given the logits $\mathbf{z}_t \in \mathbb{R}^V$ at decoding step $t$, we first compute the probability distribution via softmax:

$$p_t(i) = \frac{\exp(z_{t,i})}{\sum_{j=1}^{V} \exp(z_{t,j})}, \quad i = 1, \ldots, V, \tag{11}$$

where $V$ is the vocabulary size. The entropy of this distribution is

$$H(p_t) = -\sum_{i=1}^{V} p_t(i) \log p_t(i). \tag{12}$$

To make entropy values comparable across vocabularies of different sizes, we normalize by the maximum entropy $\log V$:

$$\hat{H}(p_t) = \frac{H(p_t)}{\log V}, \qquad \hat{H}(p_t) \in [0, 1]. \tag{13}$$

Here, $\hat{H}(p_t) \approx 0$ indicates highly confident predictions, while $\hat{H}(p_t) \approx 1$ corresponds to high uncertainty.

# D    APPENDIX: DERIVATION OF THE SOFTCFG DEVIATION BOUND

In this appendix, we derive the bound on the deviation of SoftCFG from vanilla CFG, given by:

$$\|\Delta_t^{\text{context}}\| \leq L_t \cdot \sum_{i=1}^{t-1}(1 - w_i)\|\mathbf{v}_i^{\text{uncond}}\|, \tag{14}$$

where $\Delta_t^{\text{context}} = \tilde{\mathbf{z}}_t^{\text{uncond, pertcontext}} - \mathbf{z}_t^{\text{uncond}}$ is the context-aware perturbation, and $L_t$ is the Lipschitz constant of the model $f_\theta$ with respect to the value cache.

## D.1    PERTURBED VALUE CACHE

In SoftCFG, the unconditional value cache $\mathbf{V}_{<t}^{\text{uncond}} = [\mathbf{v}_1^{\text{uncond}}, \mathbf{v}_2^{\text{uncond}}, \ldots, \mathbf{v}_{t-1}^{\text{uncond}}]$ is perturbed as:

$$\tilde{\mathbf{V}}_{<t}^{\text{uncond}} = \mathbf{W}_{<t} \odot \mathbf{V}_{<t}^{\text{uncond}}, \tag{15}$$

where $\mathbf{W}_{<t} = \text{diag}(w_1, w_2, \ldots, w_{t-1})$, $w_i = 1 - p_{\max}(x_i) \in [0, 1]$, and $\odot$ denotes element-wise scaling. Thus:

$$\tilde{\mathbf{v}}_i^{\text{uncond}} = w_i \cdot \mathbf{v}_i^{\text{uncond}}, \quad i = 1, 2, \ldots, t - 1. \tag{16}$$

The unconditional logits are computed as:

$$\mathbf{z}_t^{\text{uncond}} = f_\theta(\mathbf{K}_{<t}^{\text{uncond}}, \mathbf{V}_{<t}^{\text{uncond}}, x_{t-1}, \emptyset), \tag{17}$$

$$\tilde{\mathbf{z}}_t^{\text{uncond, pertcontext}} = f_\theta(\mathbf{K}_{<t}^{\text{uncond}}, \tilde{\mathbf{V}}_{<t}^{\text{uncond}}, x_{t-1}, \emptyset). \tag{18}$$

The deviation $\Delta_t^{\text{context}}$ arises from the perturbation $\tilde{\mathbf{V}}_{<t}^{\text{uncond}} - \mathbf{V}_{<t}^{\text{uncond}}$.

## D.2    LIPSCHITZ CONTINUITY

We assume that $f_\theta$ is $L_t$-Lipschitz continuous with respect to the value cache:

$$\|\tilde{\mathbf{z}}_t^{\text{uncond, pertcontext}} - \mathbf{z}_t^{\text{uncond}}\| \leq L_t \cdot \|\tilde{\mathbf{V}}_{<t}^{\text{uncond}} - \mathbf{V}_{<t}^{\text{uncond}}\|. \tag{19}$$

This assumption is justified by the structure of Transformer models, which consist of multi-head attention (MHA), feed-forward networks (FFN), residual connections, and LayerNorm:

- **Attention Mechanism**: The attention output is computed as $\text{Attention}(\mathbf{Q}, \mathbf{K}, \mathbf{V}) = \text{softmax}\left(\frac{\mathbf{Q}\mathbf{K}^T}{\sqrt{d_k}}\right)\mathbf{V}$. The softmax function is smooth with a Lipschitz constant of 1, as its Jacobian is bounded ($\left|\frac{\partial \text{softmax}_i}{\partial x_j}\right| \leq 1$). The value cache $\mathbf{V}$ contributes linearly, and assuming bounded key and query matrices, the attention mechanism is Lipschitz continuous.
- **Feed-Forward Network**: The FFN, $\text{FFN}(\mathbf{x}) = \sigma(\mathbf{W}_1\mathbf{x} + \mathbf{b}_1)\mathbf{W}_2 + \mathbf{b}_2$, uses activation functions like ReLU or GELU, which are Lipschitz continuous.
- **Residual Connections and LayerNorm**: Residual connections ($\mathbf{x} + \text{Attention}(\mathbf{x})$) and LayerNorm have bounded gradients.
- **Composition**: The composition of Lipschitz continuous functions (attention, FFN, LayerNorm, and final linear projection) is itself Lipschitz continuous.

Form Eq. 19, we have:

$$\|\Delta_t^{\text{context}}\| \leq L_t \cdot \|\tilde{\mathbf{V}}_{<t}^{\text{uncond}} - \mathbf{V}_{<t}^{\text{uncond}}\|. \tag{20}$$

## D.3    VALUE CACHE DIFFERENCE

The norm is:

$$\|\tilde{\mathbf{V}}_{<t}^{\text{uncond}} - \mathbf{V}_{<t}^{\text{uncond}}\| = \left\|\sum_{i=1}^{t-1}(1 - w_i)(-\mathbf{v}_i^{\text{uncond}})\right\|. \tag{21}$$

Applying the triangle inequality:

$$\left\| \sum_{i=1}^{t-1} (1 - w_i)(-\mathbf{v}_i^{\text{uncond}}) \right\| \leq \sum_{i=1}^{t-1} \left\| (1 - w_i)(-\mathbf{v}_i^{\text{uncond}}) \right\|. \tag{22}$$

Since $1 - w_i \geq 0$:

$$\left\| (1 - w_i)(-\mathbf{v}_i^{\text{uncond}}) \right\| = (1 - w_i)\|\mathbf{v}_i^{\text{uncond}}\|. \tag{23}$$

Thus:

$$\|\tilde{\mathbf{V}}_{<t}^{\text{uncond}} - \mathbf{V}_{<t}^{\text{uncond}}\| \leq \sum_{i=1}^{t-1} (1 - w_i)\|\mathbf{v}_i^{\text{uncond}}\|. \tag{24}$$

### D.4 FINAL BOUND

Combining the Lipschitz condition in Eq. 20 and the value cache difference in Eq. 24:

$$\|\Delta_t^{\text{context}}\| \leq L_t \cdot \sum_{i=1}^{t-1} (1 - w_i)\|\mathbf{v}_i^{\text{uncond}}\|. \tag{25}$$

Thus, the deviation between SoftCFG and vanilla CFG is:

$$\|\mathbf{z}_t^{\text{SoftCFG}} - \mathbf{z}_t^{\text{CFG}}\| = \gamma \cdot \|\Delta_t^{\text{context}}\| \leq \gamma \cdot L_t \cdot \sum_{i=1}^{t-1} (1 - w_i)\|\mathbf{v}_i^{\text{uncond}}\|. \tag{26}$$

This bound depends on the cumulative perturbation $\sum_{i=1}^{t-1}(1 - w_i)$, which grows with sequence length $t$, especially when many tokens have high confidence ($w_i \to 0$). If $\sum_{i=1}^{t-1}(1 - w_i)$ were bounded, the deviation would be $O(\gamma)$, ensuring SoftCFG remains within a controlled trust region. However, without Step Normalization, this sum can become large, leading to excessive deviation and potential degeneration of the unconditional branch, as discussed in Sec. 2.2.

## E APPENDIX: LIMITATIONS AND FUTURE WORK

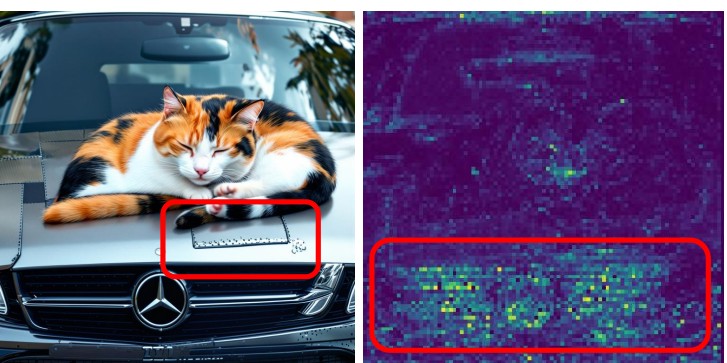

"A calico cat, sporting a patchwork of orange, black, and white fur, is comfortably nestled on top of a sleek Mercedes Benz. The luxury vehicle, with its emblem glinting in the light, is stationary, possibly in a quiet residential area. The cat, with its eyes gently shut, appears to be in a state of serene repose, oblivious to the world around it."

Figure 12: Failure case of SoftCFG. The prompt describes a calico cat on a Mercedes, but the "patchwork" attribute is wrongly applied to the car. Confidence focuses on the vehicle rather than the cat, leading to semantic drift.

While SoftCFG improves the stability and effectiveness of classifier-free guidance in visual AR models, several limitations remain.

**First, the current instantiation relies on token confidence as the weighting function.** In complex cases, the confidence distribution may collapse onto a single object, leading to concept drift where generation is overly biased toward that object. Figure 12 illustrates a representative failure mode of SoftCFG. Given the input prompt *"A calico cat, sporting a patchwork of orange, black, and white*

*fur, is comfortably nestled on top of a sleek Mercedes Benz...*", the model generates an image where the "patchwork" attribute is incorrectly transferred to the car rather than the cat. The corresponding confidence map further shows that high-confidence regions are concentrated on the vehicle, while the cat receives low confidence. As a result, SoftCFG amplifies the vehicle features and neglects the cat, leading to semantic drift. This highlights that SoftCFG is sensitive to the accuracy of the confidence estimates, which current AR models cannot yet provide reliably.

**Second, is StepNorm too strict for SoftCFG?** While step normalization effectively prevents the deviation of SoftCFG from exploding, its strict renormalization may also be overly restrictive. By enforcing $\sum_{i=1}^{t-1}(1 - \hat{w}_i) = 1$ at every step, the perturbation budget is capped uniformly across sequence lengths. This implies that no matter how long the context grows, the total perturbation injected into the unconditional branch remains equivalent to at most a single token's information. From another perspective, step normalization can be seen as discarding the cumulative contribution of multiple tokens and retaining only one token's worth of perturbation at each step. As a result, while it stabilizes long-horizon generation, its advantage in later steps may diminish compared to the unnormalized variant, see the orange line in Fig 7, where a larger perturbation budget could potentially provide stronger guidance.

Future work could address these limitations: (1) a promising direction is to incorporate auxiliary perceptual models (such as DINOV3 Siméoni et al. (2025)) to score token-level semantics, yielding more robust confidence signals for guidance. (2) Although step normalization mitigates degeneration in long sequences, extremely long-horizon generation (*e.g.*, high-resolution images or video) may still suffer from diminished guidance, calling for more adaptive normalization schemes. (3) Our study focuses primarily on class-conditional and text-to-image generation; broader applications such as multi-modal AR tasks, video synthesis, and reinforcement learning from human feedback (RLHF) remain unexplored.

# F   APPENDIX: LLM USAGE

During the preparation of this work, we made limited use of large language models (LLMs), specifically GPT, to assist with non-scientific tasks. These included (i) generating and formatting auxiliary visualization code (*e.g.*, plotting scripts for ablation studies) and (ii) refining the writing style of the manuscript through grammar correction and phrasing suggestions. All scientific ideas, theoretical formulations, experimental design, and analysis were solely developed and validated by the authors.

