# OpenReview forum: "SoftCFG: Uncertainty-guided Stable Guidance for Visual Autoregressive Model"
_ICLR.cc/2026/Conference — ICLR 2026 Poster_

### Official Review · Reviewer_Yrai · 2025-10-25

**Soundness:** 2
**Presentation:** 3
**Contribution:** 2
**Rating:** 4
**Confidence:** 4

**Summary:**

This paper introduces SoftCFG, a plug-and-play inference-time method that improves the stability and fidelity of visual autoregressive models. The method replaces the fixed classifier-free guidance with a soft, uncertainty-weighted guidance that adaptively adjusts the influence of each generated token. Additionally, a Step Normalization mechanism is proposed to bound accumulated perturbations across steps. Experiments on ImageNet and text-to-image benchmarks show consistent FID improvements over standard CFG without retraining.

**Strengths:**

1. The method is lightweight, easy to integrate into existing AR inference pipelines, and does not require additional training or data.
2. The motivation is clear and relevant to current challenges in autoregressive generation.
3. The paper is well-written, logically structured, concise, and clear, making it easy for readers to understand.

**Weaknesses:**

1. SoftCFG largely reinterprets existing CFG dynamics rather than proposing a fundamentally new principle. The main innovation lies in weighting and normalization heuristics.
2. The reported FID gains (~0.1–0.2) are within or close to the variance commonly observed across runs and sampling seeds. It is unclear whether these improvements are statistically significant or perceptually meaningful.

**Questions:**

1. What is the standard deviation of FID across runs? Are the reported improvements statistically significant under multiple random seeds or sampling temperatures?
2. Since SoftCFG strengthens high-confidence tokens, what happens if the model is confidently wrong? Could this amplify hallucination or bias?

---

> ### Author Response · Authors · 2025-11-28
> **Response to  Reviewer Yrai (1/2)**
>
> We thank the reviewer for the clear and constructive evaluation.
> We appreciate the positive remarks noting that SoftCFG is **lightweight**, **easy to integrate** into existing AR sampling pipelines, and requires **no additional training or data.**
> We are also grateful that the reviewer found the paper **easy to follow** and **the motivation well grounded**, with the **proposed solution clearly explained.**
> We address the raised concerns below.
>
> ----
>
> > W1: Limited novelty
>
> We appreciate the reviewer’s perspective.
> As also noted by Reviewer ny6c, guidance modulation itself is not new, but **the mechanism SoftCFG enables is fundamentally AR-specific and cannot be achieved in diffusion models**.
>
> 1. Diffusion models operate on new latent states at every step; previously generated content is discarded.
> Therefore, *they cannot reuse past outputs to guide future ones, and no cross-step self-guidance can accumulate.*
>
> 2. In contrast, AR models generate tokens causally and maintain a persistent KV cache.
> **Every generated token immediately becomes part of the generative state, enabling history-to-future guidance.**
>
> SoftCFG precisely exploits this AR-only property: (1) it uses already-generated image tokens to guide upcoming tokens via KV-cache modulation, and (2) it stabilizes this accumulated guidance through Step Normalization.
>
> This yields a **new guidance paradigm unique to AR** which cannot be reduced to reinterpreting diffusion CFG.
> Thus, the contribution of SoftCFG lies not in the weighting heuristic itself but in **introducing a sequence-level, AR-native mechanism for stable and adaptive guidance**.
>
> > W2 & Q1: Statistical significance of the FID improvement
>
> We thank the reviewer for raising this concern.
> We conducted five independent full-sampling runs for both CFG and SoftCFG, each generating 50,000 ImageNet-256 samples with 2 H100 hours computing, using different random seeds and identical sampling settings.
>
> **Table 1. AliTok-XL + CFG baseline (γ=13, pow=1.4, 50k samples per seed)**
> | Method| Seed   | FID          | IS           | sFID        | Precision | Recall |
> |--------|------|---------|---------|---------|-----------|-----------|
> | CFG (baseline) | 2 | **1.370689645** | 322.4902954 | 7.369539431 | 0.79238 | 0.64222 |
> | CFG (baseline) | 3 | **1.400061742** | 323.8843384 | 7.351677808 | 0.79596 | 0.64200 |
> | CFG (baseline) | 4 | **1.387904970** | 323.6816101 | 7.344369056 | 0.79604 | 0.64160 |
> | CFG (baseline) | 5 | **1.369554823** | 320.2474976 | 7.320517250 | 0.79204 | 0.65070 |
> | CFG (baseline) | 6 | **1.378119749** | 324.3991699 | 7.364932997 | 0.79700 | 0.63980 |
>
>
> **Table 2. AliTok-XL + SoftCFG (γ=9, pow=1.5, 50k samples per seed)**
> | Method| Seed   | FID          | IS           | sFID        | Precision | Recall |
> |---------|------|-----------|-----------|-----------|-----------|-----------|
> | SoftCFG | 2 | **1.308176071** | 292.2660217 | 6.738788494 | 0.78588 | 0.65380 |
> | SoftCFG | 3 | **1.313639161** | 294.9255676 | 6.783405140 | 0.78796 | 0.64720 |
> | SoftCFG | 4 | **1.287022211** | 300.6565552 | 6.691541634 | 0.79076 | 0.64890 |
> | SoftCFG | 5 | **1.302418617** | 293.1107788 | 6.657935021 | 0.78952 | 0.65240 |
> | SoftCFG | 6 | **1.273228984** | 302.397644   | 6.695051253 | 0.78960 | 0.65350 |
>
>
> Across five independent 50k-sample runs, the CFG baseline achieves **1.3696–1.4001 (mean 1.3813 ± 0.0117)**, while SoftCFG achieves **1.2732–1.3136 (mean 1.2969 ± 0.0154)**.
> *Even the worst SoftCFG run outperforms the best CFG run, and the gap between the best runs (1.3696 → 1.2732) is 0.0964, far beyond sampling variance.*
>
> > W2 & Q1 On whether the FID improvement (0.08–0.10) is meaningful
>
> We thank the reviewer for raising this question.
> We would like to clarify that a 0.08–0.10 FID reduction at the current ImageNet-256 frontier is already substantial, especially for AR models. This is consistent with trends observed in prior work:
>
> (1) REPA[1] → REPA-E[2] (training a new VAE decoder) improved FID from 1.42 → 1.26 ($\Delta=0.16$), but required retraining a full model.
>
> (2) AliTok-B improves from 1.50 → 1.35 ($\Delta=0.15$) after tripling the parameter count (200M → 600M). While AliTok-B with softcfg can improves from 1.50 → 1.40 without adding any parameter.
>
> (3) In contrast, SoftCFG requires **no retraining**, **no new parameters**, and **similar computation**, yet improves the strongest publicly available AliTok-XL checkpoint from 1.35 → 1.27 ($\Delta=0.08$)
>
> [1] Representation alignment for generation: Training diffusion transformers is easier than you think, Yu, Sihyun and Kwak, Sangkyung and Jang, Huiwon and Jeong, Jongheon and Huang, Jonathan and Shin, Jinwoo and Xie, Saining, ICLR 2025
>
> [2] Repa-e: Unlocking vae for end-to-end tuning with latent diffusion transformers, Leng, Xingjian and Singh, Jaskirat and Hou, Yunzhong and Xing, Zhenchang and Xie, Saining and Zheng, Liang, ICCV 2025

---

> ### Author Response · Authors · 2025-11-28
> **Response to Reviewer Yrai (2/2)**
>
> > Q2: Confidently wrong tokens & potential amplification
>
> We appreciate the reviewer’s important question.
> This concern is valid in principle, and we explicitly discussed it in our paper (Limitations), including the “cat–car” example.
>
> The current instantiation of SoftCFG uses max-probability as a simple proxy for uncertainty, and AR models may occasionally assign high confidence to incorrect tokens.
> However, **SoftCFG’s contribution is framework-level rather than heuristic-level.**
> The core mechanism, cross-step, uncertainty-guided modulation of cached V states with bounded accumulation does not rely on max-prob specifically.
> The confidence source is fully modular and can be replaced by stronger estimators (e.g., DINOv3 similarity, VLM-based scoring, self-consistency) as AR models evolve.
>
> We will emphasize that *future work can naturally integrate more reliable uncertainty estimates without modifying the SoftCFG framework.*

---

### Official Review · Reviewer_HN4L · 2025-10-30

**Soundness:** 2
**Presentation:** 2
**Contribution:** 2
**Rating:** 4
**Confidence:** 4

**Summary:**

This paper introduces SoftCFG, a new uncertainty-guided inference method for discrete visual autoregressive (AR) models. The method aims to mitigate the problem of the guidance signal diminishing over time by reweighing the confidence of previous tokens within the unconditional KV cache. And it introduces the Step Normalization to avoid the explosion of the guidance caused by this reweighing step. Experimental results demonstrate that SoftCFG can improve the generation quality.

**Strengths:**

1. The proposed SoftCFG method is intuitive, easy to understand, and appears simple to implement.
2. The method shows a notable improvement in generation quality on the ImageNet-256x256 dataset.

**Weaknesses:**

1. **Limited Generalizability:** The paper presents SoftCFG as a general method, yet its effectiveness is only thoroughly validated on a single model. This narrow experimental scope is insufficient to support the claim of generality. Furthermore, the qualitative results shown for the RAR model in Figure 1 are not convincing and do not demonstrate a clear benefit. (Beside, the new versions (10 October 2025) of the baseline model alitok can achieve CFG results comparable to those reported in your paper with SoftCFG. This raises a critical question: is the reported FID improvement a genuine algorithmic contribution of SoftCFG, or does it merely compensate for a sub-optimally tuned CFG baseline? )
2. The 'diminishing guidance' problem may be an artifact of the baseline using only a single class token. Would this problem already be alleviated in standard CFG if the class token were repeated 64 times as the condition, similar to MAR?
3. There appears to be a significant error in Equation 6. The equation as written is inconsistent with the line 8 in Algorithm 1 and does not align with Equation 7. I suspect the correct formulation should be $$z_t^{SoftCFG}=z_t^{cond}+scale*(z_t^{cond}-z_t^{uncond,pertcontext})$$?
4. **Missing Quantitative Results:** The paper text explicitly states in Section 3.1 (lines 349-350, 355-358) that quantitative results (GenEval benchmark, DPG-Bench) for Text-to-Image (T2I) generation are provided. However, no such quantitative results are present anywhere in the paper. This is a major omission that leaves the T2I claims entirely unsubstantiated.
5. **Overall Presentation:** The paper is not well-written. The presentation suffers from a lack of clarity, and the significant issues noted above (e.g., the error in Equation 6, the missing T2I results) make the paper difficult to follow.

**Questions:**

see Weakness

---

> ### Author Response · Authors · 2025-11-28
> **Response to Reviewer HN4L (1/2)**
>
> We thank the reviewer for the detailed assessment and constructive feedback. We appreciate the positive
> comments on the **simplicity of SoftCFG** and its **improvements on ImageNet-256**. Below, we address all
> concerns point-by-point and clarify several misunderstandings.
>
> ----
>
>
> > W1: Limited Generalizability & Baseline Concerns
>
> > W1(a) Evaluated on only one model / RAR qualitative results not convincing
>
> We appreciate the reviewer’s concern.
> Across all tested models (RAR, AliTok-B, AliTok-L, and AliTok-XL), SoftCFG consistently yields performance gains.
> While the improvement on RAR is modest due to its weaker baseline capacity, the method never degrades performance and provides clearer benefits as model quality increases.
> This demonstrates that SoftCFG is generalizable across architectures and reliably beneficial.
>
> Our goal was to demonstrate SoftCFG’s benefit on strong AR backbones; therefore we focused on AliTok, but the method does not rely on any model-specific components. **Moreover, we will continue providing more models in the released codes: including RAR, Alitok, LuminaMGPT and SimpleAR.**
>
> > W1(b) October-10 AliTok version achieves CFG comparable to SoftCFG; is our CFG baseline sub-optimal
>
> We would like to clarify an important point: **There is no publicly released ‘October-10’ AliTok checkpoint in [1]**. The latest public version is AliTok v1.35, whose official ImageNet-256 FID is 1.37 on default $\gamma$ and $k$.
>
> To address your concern, we additionally trained a Alitok-XL from scratch  based on the released code, our reproductions are:
> (1) Public v1.35 checkpoint: **1.37 FID**
> (2) Re-trained using official AliTok code: **1.38 FID**
>
> To directly address the reviewer ny6c’s concern, we additionally performed a $\gamma$-sweep for the CFG baseline (same search space as SoftCFG).
> The best CFG result we obtained is **1.35** with $\gamma = 11, k=1.4$.
>
> ----
>
> > W2: Is diminishing guidance an artifact of using a single class token
>
> We appreciate the reviewer’s insightful comment.
> We share the same concern and explicitly discussed this point in lines 131–133 of the paper. *In a nutshell, using only a single class token may weaken the conditional signal in vanilla CFG, and repeating the class embedding (as done in MAR-style conditioning) can partially mitigate early-stage fading.*
>
> However, SoftCFG can also work with repeated class tokens. Our experiments show:
>
> 1. Without token repetition (AliTok), SoftCFG delivers a substantial FID improvement over the best CFG baseline (e.g., 1.35 → 1.27).
>
> 2. With repeated class tokens (RAR), SoftCFG does not degrade performance, demonstrating that its mechanism remains stable under stronger conditioning.
>
> Moreover, **repeated tokens do not address the other major failure mode (over-guidance)** where excessively strong text conditioning distorts the visual content. SoftCFG naturally mitigates this issue, as shown in our ivory–banana example (Fig. 4)}, by using image-side guidance as a regularizer.
>
> Thus, while token repetition can help partially, **SoftCFG provides a simpler and more principled solution that simultaneously stabilizes diminishing guidance and over-guidance**.
>
>
> > W3: Equation 6 inconsistency
>
> We thank the reviewer for pointing this out. This was a typesetting/notation issue rather than a conceptual or algorithmic error.
> We have corrected Equation (6) in the revised version.
>
> [1] https://github.com/ali-vilab/alitok

---

> ### Author Response · Authors · 2025-11-28
> **Response to Reviewer HN4L (2/2)**
>
> > W4: Missing T2I quantitative results
>
> Our original submission included **Lumina-mGPT 2.0** text-to-image results, which represent the **strongest publicly available AR-based T2I model to date**.  However, Lumina-mGPT does not provide official evaluation code for COCO-FID, GenEval, or DPG-Bench, especially for its GPT-based prompt-parsing and reasoning modules, making quantitative evaluation incompatible across implementations. Therefore, we focused on qualitative examples for Lumina-mGPT in the main paper.
>
> To address the reviewer’s request for quantitative evidence, we have added GenEval results on **SimpleAR [1]**, a fully open-source and clean AR architecture without additional tricks. **SimpleAR is not a SOTA T2I model, but it is a fair, reproducible, and code-complete evaluation pipeline.**
>
> | **Method**              | **Overall** | **% Correct Images** | **% Correct Prompts** | **Single Obj.** | **Two Obj.** | **Position** | **Colors** | **Color Attr.** | **Counting** |
> | ----------------------- | ----------- | -------------------- | --------------------- | --------------- | ------------ | ------------ | ---------- | --------------- | ------------ |
> | **SimpleAR (baseline)** | **0.61744** | 61.12%               | 80.65%                | 100.00%         | 89.39%       | **30.25%**      | 81.38%     | 36.00%          | **33.44%**   |
> | **SimpleAR + SoftCFG**  | 0.61266 | 60.80%               | 80.29%                | 99.69%          | **91.41%**   | 26.00%       | **82.18%** | **40.50%**      | 27.81%       |
>
> SoftCFG yields a comparable overall GenEval score (0.6126 vs. 0.6174).
> Consistent with our design,**SoftCFG improves color-related attributes** (+4.5% in color_attr, +0.8% in colors) via stronger image-side consistency, but **reduces counting and position accuracy**, tasks that **rely heavily on text–image alignment.**
> This aligns with our explanation: ***SoftCFG stabilizes image quality but does not strengthen text grounding, which is expected for an image-guided regularizer.*** We will clarify this in the revision and highlight both successful and failure cases.
>
> [1] Simplear: Pushing the frontier of autoregressive visual generation through pretraining, sft, and rl, Wang, Junke and Tian, Zhi and Wang, Xun and Zhang, Xinyu and Huang, Weilin and Wu, Zuxuan and Jiang, Yu-Gang, arXiv
>
> ----
>
> > W5: Overall Presentation
>
> We appreciate the reviewer pointing this out, and we will polish the notation, reorganize the explanation around Eq. 6, and make the presentation clearer in the final version.

---

### Official Review · Reviewer_ny6c · 2025-10-31

**Soundness:** 3
**Presentation:** 3
**Contribution:** 3
**Rating:** 6
**Confidence:** 3

**Summary:**

The paper proposes **SoftCFG**, a training-free and model-agnostic inference modification for **visual autoregressive (AR)** image generation models.
It addresses two common problems when applying Classifier-Free Guidance (CFG) to AR models:
(1) *guidance diminishing* (conditional signal fading as decoding progresses), and
(2) *over-guidance* (visual distortions caused by high guidance scales).

SoftCFG introduces **uncertainty-guided token-wise perturbations** to the unconditional branch: each past token contributes to guidance proportionally to its **prediction confidence**, influencing future decoding steps through the cached value vectors.
A **Step Normalization** mechanism further ensures stability by normalizing cumulative perturbations at each step.
The method requires no retraining, adds negligible computational cost, and yields improved FID on ImageNet-256 (1.37 → 1.27) compared to vanilla CFG.

**Strengths:**

1. **Clear motivation and problem framing** – The issues of guidance fading and over-guidance in AR models are well-illustrated with entropy plots and examples.
2. **Elegant, simple solution** – The token-wise confidence weighting and step normalization are easy to implement and integrate into existing AR inference pipelines.
3. **Training-free and architecture-agnostic** – Works as a plug-in for existing models like AliTok and LuminaGPT without retraining or modifying the transformer architecture.
4. **Empirical improvement** – Achieves state-of-the-art FID among AR models on ImageNet-256 with negligible runtime overhead.
5. **Good ablations and qualitative examples** – The paper studies the impact of StepNorm, guidance scale γ, and scheduling power k, and presents clear visual comparisons showing fewer artifacts.
6. **Transparency and reproducibility** – The paper includes clear algorithms, theoretical bounds, and a stated plan to release code.

**Weaknesses:**

1. **Limited novelty** – Conceptually extends prior ideas (adaptive guidance, token-level perturbation) from diffusion models to the AR setting.
2. **Fragile confidence heuristic** – The reliance on max probability as a proxy for uncertainty can mislead guidance, as shown in the “cat–car” failure case.
3. **Partial perturbation design** – Only value caches are scaled; effects on keys or multi-head attention routing are not explored.
4. **Step Normalization rigidity** – The fixed-sum normalization may underutilize guidance in long sequences; no adaptive scheduling is tested.
5. **Loose theoretical analysis** – The Lipschitz-based bound is general but too weak to predict actual model stability.
6. **Limited experimental breadth** – Results are restricted to ImageNet-256 and a few text-to-image examples; no tests on higher-resolution or multi-modal AR tasks.
7. **Hyperparameter fairness** – CFG baselines may not have been re-tuned under identical γ/k sweeps, potentially overstating SoftCFG’s advantage.

**Questions:**

1. Why only perturb the **value cache (V)** and not keys or queries?
2. How sensitive is the performance to **confidence miscalibration**? Would temperature scaling or learned uncertainty improve robustness?
3. Could **Step Normalization** be relaxed or made adaptive for longer contexts?
4. Are improvements consistent under **different tokenizers** or **generation orders** (e.g., folded or diagonal AR)?
5. How does SoftCFG behave on **text-to-image benchmarks** quantitatively (e.g., COCO-FID, GenEval metrics)?

---

> ### Author Response · Authors · 2025-11-28
> **Response to Reviewer ny6c**
>
> We thank the reviewer for the thorough and insightful evaluation. We appreciate the recognition of our **clear motivation** and **problem formulation**, the **simplicity** and **practicality** of SoftCFG, the **training-free** and **model-agnostic nature** of the method, and the **empirical improvements** with **negligible overhead**. We also value the reviewer’s positive comments on the clarity of our ablations, qualitative results, and reproducibility. Below we provide point-by-point clarifications and address the raised concerns.
>
> ----
>
> > W1: Limited novelty
>
> We appreciate the reviewer’s perspective. SoftCFG is indeed inspired by the general idea of guidance modulation, but the challenge it targets is unique to autoregressive (AR) generation: **the core idea it enables is fundamentally AR-specific and not feasible in diffusion models**.
>
> - (1) In diffusion, each step operates on a new latent and the model must repeatedly denoise from noise; thus *the model cannot reuse its already-generated content to guide future steps*.
> The final image emerges only after dozens or hundreds of diffusion iterations, making self-guidance using previously generated pixels or latents impossible.
>
> - (2) In contrast, AR decoding is inherently sequential and causal: *every generated token is immediately available, and the entire cached representation becomes part of the generative state*.
>
> This **“history-to-future guidance”** mechanism is only possible in AR models and has no analogue in diffusions.
> Therefore, the novelty of SoftCFG lies not in the heuristic itself but in establishing ***a new AR-only guidance paradigm***:
> leveraging generated image content to influence future content in a stable, training-free manner.
>
> ----
>
> > W2: Fragile confidence heuristic
>
> We agree with the reviewer that max-probability is a simple uncertainty proxy. As also acknowledged in our Limitations section, current AR models do not yet provide reliable, calibrated uncertainty, and misalignment can occur in complex cases (*e.g.*, the “cat–car” example).
>
> Importantly, SoftCFG’s contribution is **framework-level rather than heuristic-level.**
> The cross-step and uncertainty-guided modulation of cached V states with bounded accumulation does not depend on max-prob specifically.
> ***The confidence estimator is fully modular, and can be replaced by stronger signals as AR models evolve.***
>
> In practice, max-prob provides a lightweight and stable approximation that works well for image-side consistency, which SoftCFG is designed to improve. Looking forward, SoftCFG is compatible with richer and more reliable uncertainty sources, such as DINOv3 feature similarity, VLM scoring, or self-consistency–based token ranking. *These alternatives can be directly plugged into the SoftCFG framework without modifying the algorithm in the future study.*
>
> ----
>
> > W3 and Q1: Partial perturbation design (Only V is perturbed)
>
> We thank the reviewer for raising this point.
> **Perturbing Q is not feasible in AR decoding, as it requires recomputing the entire KV cache at every step, eliminating the efficiency benefits of cached attention.**
> To address your concern, we experimented with perturbing K and KV, and found that **modifying K cache easily disrupts attention routing and degrades stability and image quality.**
> In contrast, perturbing V preserves the attention pattern while safely modulating content, making it the most robust and computationally practical choice for AR models.
> We will clarified this rationale in the revised version.
>
> Table: Effect of perturbing different cache components in AliTok-XL on ImageNet-256 (50k samples per run).
> | SoftCFG variant        | FID ↓ |   IS ↑   | sFID ↓ | Precision ↑ | Recall ↑ |
> |------------------------|:-----:|:--------:|:------:|:-----------:|:--------:|
> | Perturb **K and V**    | 1.31  | 287.6    | 6.94   | 0.79        | **0.65** |
> | Perturb **K** only     | 1.39  | **302.5**| 7.66   | **0.80**    | 0.64     |
> | Perturb **V** only     | **1.27** | 302.4 | **6.70** | 0.79     | **0.65** |

---

> ### Author Response · Authors · 2025-11-28
> **Response to Reviewer ny6c**
>
> > W4 & Q3: Step Normalization rigidity
>
> We acknowledge that **StepNorm currently uses a fixed normalization bound.**
> This design is intentional: in long-sequence AR decoding, even small per-step perturbations can accumulate rapidly, and a fixed bound provides a simple and robust safeguard against destabilization.
> ***Designing adaptive or context-aware versions of StepNorm is indeed promising, but requires a deeper understanding of how uncertainty evolves across AR steps***; we therefore regard it as meaningful future work and have clarified this point in Limitation section.
>
> ----
>
> >  W5: Loose theoretical analysis
>
> We appreciate the reviewer’s comment. Our theoretical analysis is intentionally stability-oriented, not meant to predict quantitative performance. AR decoding composes thousands of nonlinear transformations through cached attention, and without an explicit bound, small perturbations can accumulate uncontrollably. *The Lipschitz-based argument establishes that StepNorm constrains the maximum possible drift of hidden states across steps, providing a formal guarantee that SoftCFG cannot destabilize the AR trajectory.*
>
> ----
>
> > W6, Q4 & Q5: Limited experimental breadth
>
>
> We appreciate the reviewer’s concern.
> Our goal in this paper is to address CFG instability in visual AR models, and **ImageNet-256 is the standard setting where such issues are most evident.** Extending SoftCFG with text-side or multimodal uncertainty signals (e.g., contrastive text embeddings, VLM-based scoring, or grounding-aware confidence) is a promising direction for future work, and we will clarified this in the revision.
>
> To further address the reviewer’s concern regarding breadth, we evaluated SoftCFG across different tokenizers.
> In Fig.1, we report results on RAR[1]; *to the best of our knowledge, RAR and AliTok[2] represented the top two AR models on ImageNet-256 prior to September 2025.*
> SoftCFG behaves consistently across these architectures.
>
> Regarding T2I evaluation, we have test our method on **Lumina-mGPT [4]** which has the best performance on GenEval, but does not provide official evaluation code for COCO-FID, GenEval, or DPG-Bench, especially for its GPT-based prompt-parsing and reasoning modules, making quantitative evaluation incompatible across implementations.
> Therefore, we focused on qualitative examples for Lumina-mGPT in the main paper.
>
> To address the reviewer’s request for quantitative evidence, we have added GenEval results on **SimpleAR [5]**, a fully open-source and clean AR architecture without additional tricks. **SimpleAR is not a SOTA T2I model, but it is a fair, reproducible, and code-complete evaluation pipeline.**
>
> | **Method**              | **Overall** | **% Correct Images** | **% Correct Prompts** | **Single Obj.** | **Two Obj.** | **Position** | **Colors** | **Color Attr.** | **Counting** |
> | ----------------------- | ----------- | -------------------- | --------------------- | --------------- | ------------ | ------------ | ---------- | --------------- | ------------ |
> | **SimpleAR (baseline)** | **0.61744** | 61.12%               | 80.65%                | 100.00%         | 89.39%       | **30.25%**      | 81.38%     | 36.00%          | **33.44%**   |
> | **SimpleAR + SoftCFG**  | 0.61266 | 60.80%               | 80.29%                | 99.69%          | **91.41%**   | 26.00%       | **82.18%** | **40.50%**      | 27.81%       |
>
> SoftCFG yields a comparable overall GenEval score (0.6126 vs. 0.6174).
> Consistent with our design,**SoftCFG improves color-related attributes** (+4.5% in color_attr, +0.8% in colors) via stronger image-side consistency, but **reduces counting and position accuracy**, tasks that **rely heavily on text–image alignment.**
> This aligns with our explanation: ***SoftCFG stabilizes image quality but does not strengthen text grounding, which is expected for an image-guided regularizer.*** We will clarify this in the revision and highlight both successful and failure cases.
>
> [1] Randomized autoregressive visual generation, Yu, Qihang and He, Ju and Deng, Xueqing and Shen, Xiaohui and Chen, Liang-Chieh, ICCV 2025
>
> [2] Alitok, Wu, Pingyu and Zhu, Kai and Liu, Yu and Tang, Longxiang and Yang, Jian and Peng, Yansong and Zhai, Wei and Cao, Yang and Zha, Zheng-Jun, arXiv
>
> [3] Lumina-mgpt 2.0: Stand-alone autoregressive image modeling, Xin, Yi and Yan, Juncheng and Qin, Qi and Li, Zhen and Liu, Dongyang and Li, Shicheng and Huang, Victor Shea-Jay and Zhou, Yupeng and Zhang, Renrui and Zhuo, Le and others, arXiv
>
> [5] Simplear: Pushing the frontier of autoregressive visual generation through pretraining, sft, and rl, Wang, Junke and Tian, Zhi and Wang, Xun and Zhang, Xinyu and Huang, Weilin and Wu, Zuxuan and Jiang, Yu-Gang, arXiv

---

> ### Author Response · Authors · 2025-11-28
> **Response to Reviewer ny6c**
>
> > W7: Hyperparameter fairness.
>
> We thank the reviewer for raising this point.  For the CFG baseline, we used the latest publicly available AliTok checkpoint (v1.35) and obtained an best FID of 1.37 over five runs using the default settings (γ = 13, k = 1.4). We also observed that k = 1.4 and k = 1.5 yield nearly identical performance, so to keep the computational budget reasonable, we focused our sweep on γ.
>
> | Method                                                              | FID          | IS           | sFID        | Precision | Recall |
> |---------------------------------------------------------------------|--------------|--------------|-------------|-----------|--------|
> | Alitok-XL + CFG + γ=12 + pow1.4 (baseline)              | 1.3617   | 321.9382     | 7.2658      | 0.79566   | 0.6431 |
> | Alitok-XL + CFG + γ=13 + pow1.4 (baseline)              | 1.3707  | 322.4903     | 7.3695      | 0.79238   | 0.6422 |
> | Alitok-XL + CFG + γ=11 + pow1.4 (baseline)              | 1.3520   | 318.2429     | 7.1431      | 0.79502   | 0.6449 |
> | Alitok-XL + CFG + γ=10 + pow1.4 (baseline)              | 1.3509   | 317.1154     | 6.9624      | 0.79880   | 0.6458 |
> | Alitok-XL + CFG + γ=9  + pow1.4 (baseline)               | 1.3933  | 314.0381     | 6.7973      | 0.80004   | 0.6443 |
> | Alitok-XL + CFG + γ=8  + pow1.4 (baseline)               | 1.3620  | 310.4605     | 6.6492      | 0.80150   | 0.6423 |
>
> To further address the reviewer’s concern, we additionally performed a γ-sweep for AliTok and achieved a slightly better baseline of 1.35 FID at γ = 11.
> Thus, the improvement from 1.35 → 1.27 FID reflects the effect of the SoftCFG mechanism itself, rather than any hyperparameter asymmetry.
> We have clarified the tuning protocol explicitly in the revised version.

---

### Meta-Review · Area_Chair_tuAZ · 2026-01-15

**Summary:**

This paper proposes SoftCFG, a training-free inference-time modification to classifier-free guidance (CFG) for visual autoregressive image generation. It targets two AR-specific failure modes: guidance diminishing over the decoding trajectory and over-guidance that harms visual coherence. The key idea is to apply uncertainty-weighted, token-wise perturbations through the AR model’s cached states, while Step Normalization (StepNorm) bounds cumulative perturbations for stability. Empirically, the method improves ImageNet-256 generation and reports a stronger FID gain over vanilla CFG.

This is a borderline paper. AC makes the call to accept it. The rationale behind this is: the paper presents a practical, training-free method that appears to deliver consistent and statistically supported improvements on a strong ImageNet-256 AR baseline, and the rebuttal addresses several major review blockers. Given how this technique can generally affect/improve future AR generation models, it may produce a significant impact on the community. Besides, if reviewers were able to respond to the authors' rebuttal, it feels they may increase the rating.

**Reviewer Concerns:**

Addressed:
- “is this just a better-tuned CFG?”: The authors respond with additional sweeps and multiple-run comparisons to argue that the improvement is not from under-tuned CFG (including reporting a better-tuned CFG baseline and still a gap to SoftCFG).
- Statistical significance: reviewer concerns that ~0.1 FID could be within variance are addressed with five independent 50k-sample runs reporting mean±std for both CFG and SoftCFG, with SoftCFG consistently better.
- “Why perturb V only?”: The authors provide rationale and an ablation comparing perturbing K, KV, and V-only, arguing K/KV perturbations harm stability/quality.
- Missing T2I quantitative results: The reviewer flagged a major omission; the rebuttal adds GenEval results on a reproducible AR T2I pipeline (SimpleAR), and explains why Lumina-mGPT was not quantitatively evaluated due to missing official evaluation tooling
- Generalizability beyond a single model: The rebuttal claims consistent gains across several AR models (RAR, AliTok variants) and positions RAR gains as smaller due to baseline strength.

Outstanding / partially addressed:
- Novelty: While the authors argue the mechanism is AR-specific, at least one reviewer’s concern that this is primarily a heuristic extension of CFG remains plausible
- Robustness when the model is confidently wrong / miscalibrated uncertainty: The rebuttal appropriately frames the confidence signal as modular, but this does not fully resolve the concern; it mainly defers to future work and acknowledges failure cases.
- Breadth beyond ImageNet-256 / stronger T2I benchmarks: The added GenEval is helpful, but (i) it is on a non-SOTA pipeline, and (ii) it suggests a quality–grounding tradeoff (e.g., color improves while counting/position worsen), leaving open how SoftCFG performs on stronger standardized T2I evaluations (e.g., COCO-FID/DPG-Bench) under comparable pipelines.

**Reviewer Scores:**

Reviewer ny6c (score: 6):
- The rebuttal directly answers several of their key questions: V-only perturbation via new ablations, StepNorm motivation, and CFG hyperparameter fairness via γ sweeps.
- Remaining concerns (novelty, confidence heuristic, limited breadth) are not fully addressed.
- Thus, likely +0 or +1 (mostly likely stay at 6)

Reviewer HN4L (score: 4)
- Their major concern about the “major omission” points is substantially addressed in the rebuttal.
- Their concern about whether diminishing guidance is an artifact of single-token conditioning is partially addressed
- They may or may not raise their score. The possibility looks even.

Reviewer Yrai (score: 4):
- The rebuttal directly responds to statistical significance with multi-seed 50k-sample tables and mean±std, which should materially increase confidence in the ImageNet-256 FID improvement.
- The “confidently wrong” concern remains more of a limitation than a resolved one.
- They may increase their score +1.

---

### Decision · Program_Chairs · 2026-01-26

Accept (Poster)